# The Multifaceted Functions of Lactoferrin in Antimicrobial Defense and Inflammation

**DOI:** 10.3390/biom15081174

**Published:** 2025-08-16

**Authors:** Jung Won Kim, Ji Seok Lee, Yu Jung Choi, Chaekyun Kim

**Affiliations:** 1Laboratory of Leukocyte Signaling Research, Department of Pharmacology, College of Medicine, Inha University, Incheon 22212, Republic of Korea; kjw3791@inha.edu (J.W.K.); jiseok@inha.edu (J.S.L.); chldbwjd16@inha.edu (Y.J.C.); 2BK21 Program in Biomedical Science & Engineering, Inha University, Incheon 22212, Republic of Korea

**Keywords:** lactoferrin, infection, antimicrobial defense, inflammation, oxidative stress, cytokine

## Abstract

Lactoferrin (Lf) is a multifunctional iron-binding glycoprotein of the transferrin family that plays a central role in host defense, particularly in protection against infection and tissue injury. Abundantly present in colostrum, secretory fluids, and neutrophil granules, Lf exerts broad-spectrum antimicrobial activity against bacteria, viruses, fungi, and parasites. These effects are mediated by iron sequestration, disruption of microbial membranes, inhibition of microbial adhesion, and interference with host–pathogen interactions. Beyond its antimicrobial functions, Lf regulates pro- and anti-inflammatory mediators and mitigates excessive inflammation. Additionally, Lf alleviates oxidative stress by scavenging reactive oxygen species and enhancing antioxidant enzyme activity. This review summarizes the current understanding of Lf’s biological functions, with a particular focus on its roles in microbial infections, immune modulation, oxidative stress regulation, and inflammation. These insights underscore the therapeutic promise of Lf as a natural, multifunctional agent for managing infectious and inflammatory diseases and lay the groundwork for its clinical application in immune-related disorders.

## 1. Introduction

Lactoferrin (Lf) is an iron-binding glycoprotein that shows a high degree of sequence homology across species, with approximately 78% of the human Lf sequence being identical to that of bovine Lf [1]. It is present in various secretory fluids, including milk, saliva, and tears, as well as secondary granules of neutrophils [2]. In humans, Lf is most abundant in colostrum (~7 g/L), while mature milk (≥28 days lactation) contains around 2 g/L [3,4].

Human Lf consists of approximately 700 amino acids, has a molecular weight of ~80 kDa, and is folded into two globular lobes designated the N- and C-lobes [5,6,7,8]. The N-lobe spans amino acids 1–332, and the C-lobe includes amino acids 344–703, which are connected by a flexible α-helix linker between amino acids 333 and 343. There are three isoforms of Lf: Lf-α is the iron-binding isoform, whereas Lf-β and -γ have ribonuclease activity, although they do not bind iron [7,9]. Each lobe of Lf can bind one ferric iron (Fe^3+^), allowing the molecule to carry up to two iron atoms. Based on iron saturation, Lf exists as hololactoferrin (Holo-Lf, >85% iron saturation) or apolactoferrin (Apo-Lf, <5% iron saturation). The natural (native) Lf typically shows 10−20% iron saturation [10,11,12]. In addition to Fe^3+^ and Fe^2+^, Lf can also bind other metal ions, including Cu^2+^, Mn^2+^, and Zn^2+^ [13]. Apo-Lf adopts an open conformation, while Holo-Lf has a closed structure that confers greater resistance to proteolysis [5]. These structural differences contribute to distinct functional properties (Figure 1) (Table 1). Although Holo-Lf is more stable, Apo-Lf exhibits stronger antimicrobial and immunomodulatory activities, largely due to its ability to sequester iron from the environment, thereby depriving microbes of essential nutrients. Apo-Lf also demonstrates superior antioxidant activity compared to Holo-Lf [14].

Lf plays diverse and essential biological roles, including facilitating iron absorption and exerting antimicrobial, anti-inflammatory, antioxidant, and immunomodulatory effects [34,35,36]. These functions are mediated through interactions with a variety of receptors, such as low-density lipoprotein-related protein-1 (LRP-1/CD91/apoE receptor), chylomicron remnant receptor, intelectin-1 (omentin-1), nucleolin, toll-like receptor (TLR)2, TLR4, CXCR4, CD14, SD206, heparan sulfate proteoglycans (HSPGs), and interleukin (IL)-1 [37]. The expression of Lf receptors varies by tissue and cell type [38]. LRP-1 is expressed on monocytes, macrophages, hepatocytes, and endothelial cells, where it mediates the endocytosis of Lf, clears Lf-bound complexes, and modulates inflammatory responses [39]. Nucleolin is expressed on the surface of certain cancer cells and immune cells, where it facilitates Lf internalization. IL-1 receptors, found on intestinal epithelial cells, adipose tissue, and immune cells, promote Lf uptake, iron absorption, and contribute to the immune defense mechanism.

There is extensive evidence supporting Lf’s multifunctional roles in antimicrobial function, immune regulation, and oxidative stress mitigation. Moreover, Lf demonstrates therapeutic potential across a range of infectious and inflammatory conditions. However, a comprehensive understanding of how these diverse biological functions are linked and can be effectively leveraged for clinical application remains limited. This review aims to provide an integrated overview of Lf’s biological functions and to explore its potential as a therapeutic agent. To this end, we outline key microbial species inhibited by Lf and cytokines modulated by its activity, offering a valuable reference to inform future research and the development of Lf-based therapeutic strategies.

## 2. Lf Release from Neutrophils

Lf is synthesized by granular epithelial cells in exocrine fluids and by neutrophils. In neutrophils, Lf is stored in specialized secondary (specific) granules, which are formed during neutrophil maturation in the bone marrow [38]. Upon activation by bacterial products (e.g., lipopolysaccharide; LPS), immune complexes, chemokines, or cytokines, neutrophils initiate degranulation. The granules migrate to and fuse with either the plasma membrane or phagosomes, releasing their contents, including Lf, into the extracellular space or phagosome. Once released, Lf performs key antimicrobial functions. It sequesters iron to inhibit microbial growth, disrupts microbial membranes, and modulates immune responses by reducing excessive inflammation. Therefore, neutrophil dysfunction or impaired degranulation may lead to reduced Lf release and compromised host defense. Under normal conditions, blood levels of Lf are low (200–500 µg/L), but they can rise significantly, to as much as 200 mg/L, during infections and inflammatory responses, reflecting increased neutrophil numbers and degranulation [2,38,40,41,42,43,44]. Notably, a single million human neutrophils can release approximately 15 µg of Lf [38]. Accordingly, elevated Lf levels in body fluids can serve as biomarkers of inflammation, particularly in diseases such as inflammatory bowel disease.

## 3. Antimicrobial Activity

The antimicrobial activity of Lf has been extensively documented against a broad spectrum of pathogens, including bacteria, viruses, fungi, yeasts, and parasites [42,45]. Lf exerts its antimicrobial effects through multiple mechanisms: it binds iron with high affinity, depriving microbes of this essential nutrient; disrupts microbial membranes; and inhibits bacterial adhesion and biofilm formation. Additionally, Lf interferes with microbe–host cell interactions by binding to microbial components or host cell receptors, thereby blocking pathogen entry and colonization. Beyond its direct antimicrobial actions, Lf enhances the host immune response by stimulating immune cells and promoting cytokine production. Due to its broad-spectrum activity, low toxicity, and immunomodulatory properties, Lf is considered as a promising therapeutic candidate, particularly in the context of antibiotic resistance, adverse drug reactions, and the need for immune-supportive interventions.

### 3.1. Antibacterial Activity

The antibacterial activity of Lf is primarily attributed to its ability to sequester free iron, depriving bacteria of this essential element for growth and metabolism. In Gram-negative bacteria, Lf interacts with bacterial LPS on the outer membrane, disrupting membrane integrity and competing with CD14 for LPS binding, thereby preventing downstream activation of TLRs on immune cells [46]. In Gram-positive bacteria, Lf’s cationic nature enables it to bind to anionic surface molecules such as lipoteichoic acid, reducing surface charge and destabilizing the membrane. This disruption facilitates lysozyme access to the underlying peptidoglycan, enhancing its enzymatic effect [7,47]. Additionally, Lf may exert antibacterial effects through the generation of peroxides catalyzed by Lf-bound iron ions, leading to altered membrane permeability and bacterial cell lysis [46,48,49,50,51,52].

Lf demonstrated antibacterial activity against a wide range of Gram-negative bacteria, including [53,54], *Enterobacter* spp., *Escherichia coli*, *Haemophilus influenzae*, *Helicobacter felis*, *Helicobacter pylori*, *Klebsiella pneumoniae*, *Porphyromonas gingivalis*, *Pseudomonas aeruginosa*, *Salmonella*, and *Yersinia* spp. [55,56,57,58,59,60,61,62,63,64,65] (Table 2), as well as Gram-positive bacteria such as *Bacillus cereus*, *Listeria monocytogenes*, and *Staphylococcus aureus* [66,67] (Table 3). Lf inhibits bacterial growth by sequestering iron, a critical element for microbial metabolism, and by interacting with key bacterial components such as protein A, lysozyme, and DNA [68]. These interactions disrupt essential cellular functions and inhibit biofilm formation, particularly in *P. aeruginosa* and *S. aureus* infections, which are known for their antibiotic resistance and chronicity [21]. Moreover, Lf has been shown to enhance the efficacy of various antibiotics, including gentamicin, levofloxacin, rifampicin, clarithromycin, and clindamycin, demonstrating effects that could lower required drug doses and reduce adverse effects [61,69]. These properties position Lf as a promising adjunctive agent in the management of multidrug-resistant bacterial infections.

### 3.2. Antiviral Activity

Lf demonstrated broad-spectrum antiviral activity, as comprehensively reviewed by Eker et al. [86]. It exerts inhibitory effects against a wide array of DNA and RNA viruses, including adenoviruses, cytomegalovirus, enteroviruses, echovirus, Japanese encephalitis virus, hepatitis C virus (HCV), herpes simplex virus, influenza virus, human cytomegalovirus, human immunodeficiency virus (HIV), human norovirus, human respiratory syncytial virus, papillomavirus, poliovirus, rotavirus, Zika virus, and most notably SARS-CoV-2 [87,88,89,90,91,92,93,94,95,96,97,98,99,100,101] (Table 4).

Mechanistically, Lf binds directly to viral particles or to host cell surface receptors, including viral hemagglutinin, HSPGs, and angiotensin-converting enzyme 2, thereby preventing viral attachment, fusion, and entry [131,132,133]. This receptor-binding inhibition is a critical first step in disrupting the infection cycle, as Lf blocks the initial interaction between virus and host cells [134]. In addition, Lf interferes with later stages of viral infection, such as viral internalization (e.g., poliovirus type 1) and replication (e.g., rotavirus, HCV) [60,114,135]. Interestingly, while most studies report antiviral effects, some findings suggest that Lf may enhance adenovirus infection by promoting viral attachment to epithelial cells, highlighting the virus-specific nature of Lf activity [104]. The authors proposed that Lf facilitates adenovirus infection by binding to the coxsackievirus and adenovirus receptor (CAR), which allows entry into target cells. However, CAR is rarely expressed on the apical side of polarized cells, which initiates infection. Therefore, it seems that adenovirus uses Lf as a bridge to attach to host cells. Moreover, Lf has been shown to act synergistically with established antiviral agents, including acyclovir, ribavirin, and zidovudine, enhancing their therapeutic efficacy [69,112].

Since the emergence of COVID-19, Lf has been extensively investigated for its role in SARS-CoV-2 inhibition. Animal studies and clinical studies reported shorter symptom duration, improved recovery rates, and reduced viral loads in SARS-CoV-2 infection supplemented with Lf [99,100,101]. Notably, reduced Lf levels have been observed in the milk from mothers infected with SARS-CoV-2 [136], suggesting a potential correlation with host defense.

### 3.3. Antifungal Activity and Antiparasitic Activity

Lf exhibits notable antifungal activity through multiple mechanisms, including iron sequestration, disruption of fungal membrane integrity and increased membrane permeability, and the induction of apoptosis [137,138]. Lf has been shown to inhibit the growth of several pathogenic fungi, such as *Aspergillus fumigatus*, *Candida* spp., *Cryptococcus neoformans*, and *Trichophyton mentagrophytes* (Table 5). Moreover, Lf acts synergistically with conventional antifungal agents, including amphotericin B, fluconazole, and caspofungin [139,140]. The bioactive peptide derived from Lf, lactoferricin (positively charged N-terminal 49 residues of Lf) exhibits even greater antifungal potency than the parent protein by inserting into the fungal membrane, resulting in membrane destabilization and cell death, even at low concentrations, and is particularly effective against drug-resistant strains [141].

Lf also possesses broad antiparasitic activity, as reviewed by Anand [159] and Zarzosa-Moreno et al. [160], and targets a range of intestinal and blood-borne protozoan parasites. Its antiparasitic mechanisms parallel those observed in antibacterial and antifungal actions, involving iron deprivation, membrane disruption, and interference with host–parasite interactions. For example, Lf binds to the membrane lipids of *Entamoeba histolytica* trophozoites, causing membrane disruption and parasite death [161,162]. It also inhibits the proliferation or viability of *Babesia caballi*, *Cryptosporidium sporozoites*, *Entamoeba histolytica*, *Giardia lamblia* (by blocking adherence to host epithelial cells), *Leishmania* spp., *Plasmodium berghei*, and *Plasmodium falciparum*, where it not only suppresses parasite growth, but also acts synergistically with antimalarial drugs [36,163,164,165,166,167,168,169,170,171,172]. In addition, Lf inhibits *Toxoplasma gondii* (by suppressing intracellular growth) and *Trichomonas vaginalis* (by blocking epithelial binding), and enhances the killing of *Trypanosoma* spp. [173,174,175,176,177]. Moreover, lactoferricin shows superior antiparasitic effects compared to native Lf due to its enhanced ability to penetrate and disrupt parasite membranes. These findings again highlight the therapeutic potential of both Lf and its derivatives as adjunctive and alternative treatments for fungal and parasite infections, particularly in the face of increasing drug resistance.

## 4. Anti-Inflammatory Activity

Lf exhibits potent anti-inflammatory properties by modulating a wide range of inflammatory responses. During microbial infection, Lf neutralizes microbial components such as LPS, thereby preventing the activation of pro-inflammatory signaling pathways. It interferes with key regulators of inflammation, including mitogen-activated protein kinase (MAPK) and nuclear factor-kappa B (NF-κB), contributing to the resolution of inflammation and the restoration of immune homeostasis [178]. Lf also regulates the activity of various immune cells, such as neutrophils, macrophages, and dendritic cells. Its anti-inflammatory effects are further supported by its ability to suppress the production of pro-inflammatory cytokines, including IL-1β, IL-6, and tumor necrosis factor-alpha (TNF-α). Owing to its ability to modulate inflammation, Lf holds therapeutic potential for treating inflammatory conditions such as sepsis, inflammatory bowel disease, neuroinflammation, and respiratory infections, as well as reducing hyperoxia-induced kidney and lung injuries [69].

### 4.1. Regulation of Cytokines

Lf plays a critical role in modulating the balance between pro- and anti-inflammatory cytokines, thereby promoting immune homeostasis [179,180]. Lf has been shown to downregulate the production of pro-inflammatory cytokines, such as interferon (IFN)-γ, IL-1β, IL-2, IL-6, IL-8, and TNF-α, in various cell types, including human mononuclear cells, endometrial stromal cells, THP-1, RAW 264.7, and A549 cells. In contrast, Lf upregulates anti-inflammatory cytokines, including IL-4 and IL-10. Some studies report that Lf can induce the production of IL-6, IL-8, and TNF-α [181,182,183], suggesting that Lf may act as a mild inflammatory stimulus. Nevertheless, Lf is also capable of suppressing stimulus-induced overproduction of pro-inflammatory cytokines, indicating its broader regulatory function in preventing excessive inflammation. However, the effects of Lf on certain cytokines, including IL-6, IL-10, IL-18, and TNF-α, have been inconsistent (Table 6).

Lf activates macrophages and enhances TNF-α production [182,183], whereas it suppresses TNF-α production under conditions of chronic endometritis, pregnancy, and lung cancer [198,202,211]. Moreover, Lf differentially regulates the production of cytokines, such as IL-6, IL-10, and TNF-α, depending on the timing of treatment in LPS-induced inflammatory mice [212]. These results suggest that Lf differentially regulates inflammatory cytokines in a context-dependent manner: it promotes pro-inflammatory cytokine production under physiological conditions but suppresses them in pathological or disease states.

Oral administration of Lf has been associated with significant alterations in cytokine profiles in both humans and animal models. In these studies, Lf reduced levels of pro-inflammatory cytokines, such as INF-γ, IL-6, IL-8, macrophage migration inhibitory factor (MIF), and TNF-α [185,198,208]. Concurrently, it enhanced levels of anti-inflammatory cytokines, including IL-4 and IL-10, in mice and rats [190]. Notably, in pregnant women, Lf supplementation led to a reduction in IFN-γ, IL-1α, IL-4, IL-9, IL-15, IP-10, MCP-3, and TNF-α, while increasing levels of IL-17, fibroblast growth factor-basic (FGF-basic), granulocyte colony-stimulating factor (G-CSF), and granulocyte-macrophage colony-stimulating factor (GM-CSF) [202].

### 4.2. Mitigation of Oxidative Stress

Lf demonstrates significant potential in mitigating oxidative stress across various biological systems, primarily through its ability to sequester free iron [69,213]. By binding iron, Lf limits its availability for participation in the Haber–Weiss reaction, thereby reducing the generation of free radicals. In addition to iron sequestration, Lf directly scavenges hydroxyl radicals and can undergo oxidative self-degradation to neutralize reactive species [214]. It significantly reduces the production of reactive oxygen species (ROS) induced by a variety of oxidative stimuli, including hydrogen peroxide (H_2_O_2_), LPS, prion proteins, dexamethasone, and alcohol [215,216,217,218], in different cell types, such as human neutrophils and human mesenchymal stem cells, as well as MC3T3-E1, SH-SY5Y, A549, and AML-12 [25,213,218,219,220,221]. These findings suggest that Lf may ameliorate inflammation, at least in part, by limiting ROS production. Lf can also promote ROS production under certain conditions. For example, Holo-Lf has been shown to increase ROS levels in erythrocytes by enhancing the Fenton reaction, leading to hemolysis [222]. In addition, In *C. albicans*, Lf induced substantial ROS accumulation, triggering an apoptosis-like response that could be alleviated by antioxidants such as menadione and N-acetylcysteine [223]. These findings suggest that the antimicrobial activity of Lf may, in some cases, be mediated through ROS generation in microbial cells.

In addition to reducing ROS directly, Lf mitigates oxidative stress by enhancing the expression and activity of antioxidant enzymes, such as superoxide dismutase (SOD), catalase (CAT), glutathione (GSH), and glutathione peroxidase (GPX) [215,224,225,226] (Table 7). Lf has been shown to dose-dependently increase GSH levels in erythrocytes and restore antioxidant enzyme activity diminished by various toxic substances. For instance, it reverses acrylamide-induced reductions in CAT, GSH, and SOD activity [224], as well as hexavalent chromium-induced suppression of CAT, GSH, and SOD in rat testicular tissues [227]. Similarly, it restores dietary deoxynivalenol-induced suppression of GPX activity in mouse testes [226], and increases GSH and DPPH levels in mouse liver and SOD activity in aged mice. Furthermore, Lf overexpression in astrocytes is associated with upregulation of antioxidant enzymes such as SOD1 and GPX4 [4,225]. Therefore, Lf mitigates oxidative stress through a dual mechanism: by directly inhibiting and scavenging ROS and by enhancing the body’s oxidant defenses.

## 5. Concluding Remarks

LF exhibits broad-spectrum antimicrobial activity and plays a critical role in modulating immune responses, thereby contributing to host defense and maintaining immune homeostasis (Figure 2). Its abundance in milk, particularly in colostrum, underscores its importance in neonatal and infant immunity, while neutrophil-derived Lf continues to contribute to immune defense throughout life. In addition to its antimicrobial properties, Lf actively facilitates the resolution of inflammation by modulating the balance between pro- and anti-inflammatory cytokines and alleviating oxidative stress (Table 8). These diverse functions make Lf a promising therapeutic candidate for the prevention and treatment of infectious and inflammatory diseases, as well as for protection against hyperoxia-induced tissue injury.

Clinical applications of Lf primarily focused on conditions including anemia, hepatitis C infection, type 2 diabetes, and colorectal polyps. In pregnant women, supplementation with iron-saturated bovine Lf led to increased hemoglobin and total serum iron levels while reducing IL-6 production, indicating both hematologic and anti-inflammatory benefits [203,243,244,245,246]. Moreover, the safety and effectiveness of Lf compared to ferrous sulfate treatment have been reported [247]. In patients with chronic hepatitis C, Lf administration resulted in a decrease in HCV viral load and a reduction in serum alanine transaminase levels [117,248]. Furthermore, Lf has been shown to inhibit the growth of adenomatous colorectal polyps, suggesting its potential role in colorectal cancer prevention and as an adjunctive therapy following polyp extraction [249].

Regarding safety and toxicity, Vishwanath-Deutsch et al. reviewed evidence indicating that Lf is well-tolerated and safe in both animal and human studies [250]. Animal studies showed no significant toxicity across safety or tolerability endpoints, with no observed adverse effects even at the highest tested doses. Additionally, no studies specifically identified increased immunogenicity or allergenicity associated with Lf. Furthermore, Lf is expected to enhance drug bioavailability by encapsulating therapeutic agents and protecting them from degradation. The improved bioavailability may be particularly beneficial for treating intestinal inflammatory disorders. Lf can cross biological barriers, including the blood–brain barrier and intestinal epithelium, making it a promising vehicle for drug delivery. This property is particularly advantageous for targeting neurodegenerative diseases such as Parkinson’s disease. Conjugation of therapeutic agents with Lf can improve their solubility, stability, and targeted delivery for antimicrobial, anti-inflammatory, and chemotherapeutic agents. Therefore, Lf holds promise as a carrier for drugs and bioactive molecules.

## Figures and Tables

**Figure 1 biomolecules-15-01174-f001:**
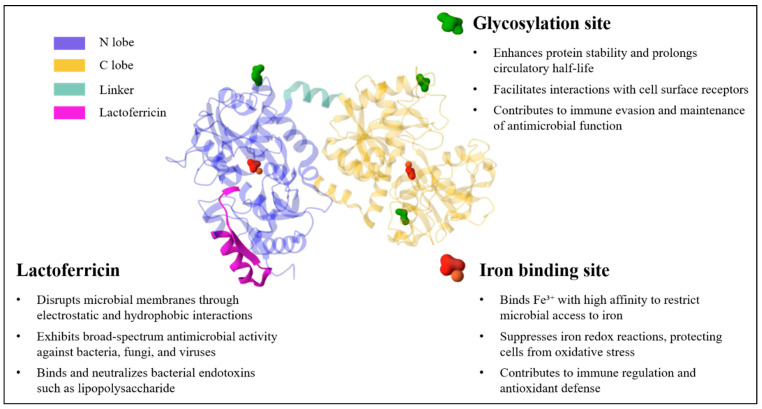
The structure and related functions of human lactoferrin. Structure visualization of lactoferrin (PDB: 1B0L) generated using Mol * (RCSB PDB, https://www.rcsb.org).

**Figure 2 biomolecules-15-01174-f002:**
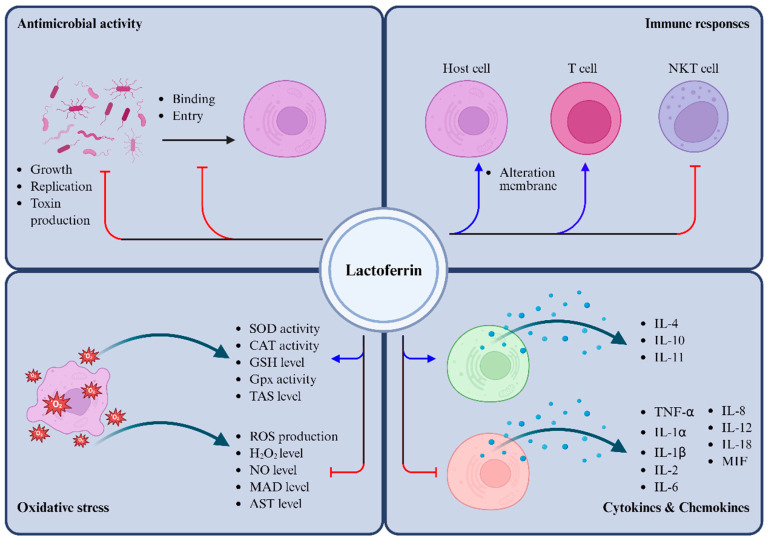
Overview of lactoferrin’s main action in host defense and inflammation. Activation is shown by blue lines and inhibition by red lines.

**Table 1 biomolecules-15-01174-t001:** Function of Apo- and Holo-lactoferrin.

Function	Apo-Lf (Iron-Free)	Holo-Lf (Iron-Bound)	Ref.
Iron scavenging	High capacity to bind iron (good at chelating iron from the environment)	Cannot bind more iron (saturated)	[15,16,17,18,19,20,21,22,23,24,25]
Interaction with bacteria	Disrupt bacterial membranes and inhibit growth	Less effective	[18,26,27]
Antimicrobial activity	Stronger: depriving pathogens of the iron they need to grow	Weaker: no longer chelates iron	[16,17,18,21,22,26]
Immunomodulation	More potent in anti-inflammatory effects	Moderate to weak	[25,28]
Stability	Less stable (more prone to degrade in acidic environments)	More stable due to an iron-induced conformational change	[20,29,30,31,32,33]

**Table 2 biomolecules-15-01174-t002:** Effect of lactoferrin on Gram-negative bacteria.

Bacteria	Host	Function and Mechanism	Ref.
*Chlamydophila psittaci*	in vitro	Inhibit attachment and entry	[70]
*Chlamydia trachomatis*	in vitro	Inhibit entry Reduce IL-6 and IL-8	[53,54]
*Enterobacter sakazakii*	in vitro	Inhibit growth	[55]
*Escherichia coli*	in vitro	Inhibit adherence	[57]
	in vitro	Impair type III secretory system	[56]
	in vitro	Inhibit growth	[67]
	in vitro	Inhibit biofilm formation	[66]
*Haemophilus influenzae*	in vitro	Inactivate colonization factors	[58]
*Helicobacter felis*	Mouse	Reverse gastritis, infection rate, and gastric surface hydrophobicity changes	[59]
*Helicobacter pylori*	Mouse	Inhibit gastric colonization and inflammation	[71]
	Mouse	Reduce bacterial load Inhibit TNF-α, IFN-γ, IL-17, COX-2 Increase IL-4, IL-10, IL-12 Regulate blood parameters Alleviate histopathological changes	[72]
	in vitro	Inhibit growth	[73]
*Klebsiella pneumoniae*	in vitro	Enhance sensitivity to antibiotics	[61]
*Porphyromonas gingivalis*	Human	Inhibit growth	[62]
*Pseudomonas aeruginosa*	in vitro	Inhibit biofilm formation	[63]
	Mouse	Decrease weight loss Inhibit growth Decrease cell infiltration Decrease MCP-1 and MIP-1	[74]
*Salmonella enterica s* erovar Typhimurium	Mouse	Decrease bacterial load in the liver and spleen Reduce hepatomegaly and splenomegaly	[64]
	Mouse	Increase survival Decrease weight loss Inhibit infection	[75]
	Mouse	Increase survival Inhibit infection Increase IgA and IgG	[76]
*Yersinia*	in vitro	Inhibit entry	[65]

**Table 3 biomolecules-15-01174-t003:** Effect of lactoferrin on Gram-positive bacteria.

Bacteria	Host	Function and Mechanism	Ref.
*Bacillus cereus*	in vitro	Inhibit growth	[77,78]
*Clostridium difficile*	in vitro	Delay growth Prevent toxin production	[79]
*Listeria monocytogenes*	in vitro	Inhibit biofilm formation	[66]
*Staphylococcus aureus*	Mouse	Decrease IL-17 and IL-1β	[80]
	in vitro	Increase IL-1β, IFN-γ, IL-2, Reduce IL-6, IL-1β, IL-12p40	[80]
	Mouse	Inhibit kidney infection	[81]
	Mouse	Increase the spleen cell number Increase IFN-γ and TNF-α Reduce IL-5 and IL-10	[82]
	in vitro	Decrease cell viability	[83]
	in vitro	Inhibit growth	[67]
*Streptococcus mutans*	in vitro	Inhibit aggregation and biofilm	[84]
	Mouse	Reduce cavity development	[85]
	in vitro	Inhibit infection	[85]

**Table 4 biomolecules-15-01174-t004:** Effect of lactoferrin on viruses.

Viruses	Host	Function and Mechanism	Ref.
Adenovirus	in vitro	Inhibit viral antigen synthesis	[102]
	in vitro	Promote binding and infection	[103,104]
Avian influenza	Mouse	Decrease weight loss Decrease IL-17, IL-22, TNF-α	[105]
Cytomegalovirus	Mouse	Reduce infection	[93]
Enterovirus E	in vitro	Inhibit virus replication	[106]
Enterovirus 71	Mouse	Increase survival	[107]
	in vitro	Inhibit infection Decrease IL-6 Increase IFN-α	[107]
	in vitro	Inhibit infection	[108]
Hepatitis B virus (HBV)	in vitro	Inhibit virus binding Inhibit infection	[109,110]
	in vitro	Inhibit growth	[111]
Hepatitis C virus (HCV)	in vitro	Inhibit entry	[112,113,114]
		Inhibit virus replication	[114]
	in vitro	Inhibit virus replication Inhibit viral ATPase/helicase	[115]
	in vitro	Inhibit entry Inhibit virus replication	[112,116]
	Human	Decrease serum ALT Decrease HCV RNA level	[117]
Herpes simplex virus	in vitro	Inhibit infection	[118]
Influenza	in vitro	Reduce infection	[119]
	in vitro	Suppress viral antigen synthesis Reduce infection	[120]
	in vitro	Inhibit cell apoptosis Inhibit DNA fragmentation Reduce caspase-3 activity	[121]
	in vitro	Inhibit virus replication	[122]
	in vitro	Inhibit infection	[123]
Mayaro virus	in vitro	Inhibit infection Inhibit entry	[124]
Rhinovirus B-14	in vitro	Reduce virus binding	[125]
Rotavirus	in vitro	Inhibit viral cytopathic effect	[60,126]
SARS-CoV-2	in vitro	Inhibit infection and replication Reduce thymic stromal lymphopoietin Upregulate TGF-β1	[127]
	in vitro	Reduce virus binding Obscure host cell receptors	[128]
	in vitro	Reduce virus binding	[98]
	Rat	Decrease TNF-α, IL-4, IL-1β, IL-6, IL-10 Increase CD4 cells Alleviate pulmonary fibrosis	[99]
	in vitro	Increase virus neutralization Inhibit virus propagation	[99]
	in vitro	Decrease virus infection	[100]
	Hamster	Decrease virus infection Alleviate pulmonary histopathological changes	[100]
	in vitro	Decrease virus infection Inhibit entry	[101]
	Mouse	Decrease IFN-γ Increase IL-1β, IL-2, IL-6, GM-CSF Increase TLR-4 and TLR-9	[129]
	in vitro	Decrease NK and NKT cells Activate CD4 cells Decrease programmed death of CD4 and CD8 cells Increase CCL5	[129]
Toscana virus	in vitro	Inhibit viral cytopathic effect	[130]

**Table 5 biomolecules-15-01174-t005:** Effect of lactoferrin on fungi.

Fungi	Host	Function and Mechanism	Ref.
*Aspergillus fumigatus*	Human	Inhibit growth Iron deprivation	[142]
	in vitro	Iron deprivation	[142,143]
	in vitro	Prevent biofilm	[140]
*Candida albicans*	Mouse	Inhibit growth Downregulate EGF1	[144]
	Mouse	Inhibit growth Increase IL-10, TNF-α, IFN-γ, MCP-1	[145]
	Galleria mellonella	Decrease fungal burden	[140]
	in vitro	Iron deprivation Interact with cell surface Alter cell membrane Alter cell membrane H^+^ ATPase	[137,139,146,147,148,149,150]
*Candida glabrata*	in vitro	Interact with cell surface Alter cell membrane	[139,148]
*Candida guilliermondii*	in vitro		[148]
*Candida krusei*	in vitro		[147,151,152]
*Candida parapsilosis*	in vitro		[148]
*Candida tropicalis*	in vitro		[148]
*Cryptococcus gattii*	in vitro	Iron deprivation	[153]
*Cryptococcus neoformans*	in vitro	Iron deprivation Alter responses to stress	[153,154]
	in vitro	Disrupt iron transport Inhibit growth	[155]
	Galleria mellonella	Inhibit growth Interact with cell surface Reduce cell and capsule size	[140]
*Saccharomyces cerevisiae*	in vitro	Regulate cell death	[156]
	in vitro	Iron deprivation	[153]
*Trichophyton mentagrophytes*	in vitro	Inhibit growth	[152]
	Guinea pig	Inhibit growth	[152]
	Guinea pig	Modulate mononuclear cell function	[157]
*Trichophyton* spp.	in vitro	Interact with cell surface Alter cell membrane	[158]

**Table 6 biomolecules-15-01174-t006:** Effect of lactoferrin on cytokine levels.

Cytokines	Stimuli	Lf Effects	Ref.
IFN-γ	Co26Lu tumor	I	[184]
	LPS	D	[185]
	*Toxoplasma gondii* cysts	D	[186]
IL-1α	NaOH	D	[187]
IL-1β	No stimulus	D	[188]
	Dextran sulfate sodium (DSS)	D	[189,190]
	Deoxynivalenol	D	[191]
	LPS	I	[192]
	LPS	D	[35,179,193,194]
	LPS + IFN-γ	D	[35]
	NaOH	D	[187]
	Thioacetamide (TAA)	D	[195]
	Trehalose 6,6′-dimycolate (TDM)	D	[196,197]
	TNF-α	D	[198]
	2, 4, 6-trinitrobenzenesulfonic acid (TNBS)	D	[199]
	*Burkholderia cenocepacia*	D	[200]
	*Prevotella intermedia*	D	[201]
IL-2	LPS	D	[179]
IL-4	No stimulus	D	[202]
	DSS	I	[190]
	TNBS	I	[199]
IL-6	No stimulus	I	[203,204]
	No stimulus	D	[188]
	CCl_4_	D	[187]
	DSS	D	[190]
	H_2_O_2_	D	[205]
	LPS	I	[181,192]
	LPS	D	[35,185,206]
	LPS + IFN-γ	D	[35]
	TAA	D	[195]
	TDM	D	[197]
	TNF-α	D	[198]
	*Chlamydia trachomatis*	D	[54]
	*Escherichia coli* HB101(pRI203)	D	[207]
	*Mycobacterium tuberculosis*	D	[196]
	*P. intermedia*	D	[201]
IL-8 (CXCL8)	No stimulus	I	[182]
	Deoxynivalenol	D	[191]
	H_2_O_2_	D	[205]
	LPS	D	[35]
	Sepsis-induced acute lung injury	D	[208]
	*C. trachomatis*	D	[54]
	*E. coli* HB101(pRI203)	D	[207]
	*P. intermedia*	D	[201]
IL-10	No stimulus	D	[209]
	Deoxynivalenol	I	[191]
	DSS	I	[190]
	LPS	D	[35,183,185]
	LPS + IFN-γ	I	[35]
	TAA	I	[195]
	TDM	I	[196]
	TNBS	I	[199]
	*T. gondii* cysts	I	[186]
IL-11	Zymosan	I	[210]
	*B. cenocepacia*	I	[200]
IL-12	LPS	I	[183]
IL-18	No stimulus	D	[188]
	Co26Lu tumor	I	[184]
	*T. gondii* cysts	D	[186]
MIF	Sepsis-induced acute lung injury	D	[208]
	*Pseudomonas aeruginosa*	D	[74]
TNF-α	No stimulus	I	[182,183]
	No stimulus	D	[198,202,211]
	Deoxynivalenol	D	[191]
	DSS	D	[189,190]
	LPS	D	[35,179,181,192,194,212]
	Sepsis-induced acute lung injury	D	[208]
	TDM	D	[197]
	TNBS	D	[199]
	*E. coli* HB101(pRI203)	D	[207]
	*M. tuberculosis*	I	[196]
	*P. intermedia*	D	[201]

D: decrease, I: increase.

**Table 7 biomolecules-15-01174-t007:** Effect of lactoferrin on oxidative stress.

**Oxidative-Stress**				
**Oxidative Stress**	**Study Model**	**Stimuli**	**LF Effects**	**Ref.**
Intracellular ROS	in vitro (A549)	Ragweed pollen extract (RWE), Glucose oxidase (Gox)	D	[25]
	in vitro (NHBE)	RWE		
	in vitro (U937)	Gox	D	[228]
	in vitro (SH-SY5Y)	PrP (106–126)	D	[216]
	in vitro (RBC)		D	[222]
	in vitro (hMSC)	H_2_O_2_	D	[213]
	in vitro (MC3T3-E1)	H_2_O_2_	D	[220]
	in vivo (hippocampus)	Age	D	[229]
	in vitro (N2a)	Ferric ammonium citrate (FAC)	D	[225]
	in vitro (AML-12)	Ethanol	D	[218]
	in vitro (FL83B)	Thioacetamide	D	[195]
	in vitro (CCD-841-CON, CCD-18co, HT29)	Lipopolysaccharide (LPS)	D	[230]
	in vitro (U937, AML-12)	LPS, H_2_O_2_, Gox	D	[231]
H_2_O_2_	in vitro (neutrophil)	LPS	D	[219]
	in vitro (A549)	RWE	D	[25]
	in vivo (BAL fluid)			
	in vivo (plasma)	Dexamethasone	D	[217]
	in vitro (HUVEC)	H_2_O_2_	D	[232]
	in vivo (serum, liver, kidney)	HgCl_2_	D	[215]
	in vitro (U937, AML-12)	LPS, H_2_O_2_	D	[231]
	in vivo (liver, heart, muscle, brain)	LPS, H_2_O_2_	D	[231]
Nitric oxide (NO)	in vivo (liver)	Bleomycin	D	[233]
	in vivo (liver)	LPS	D	[234]
	in vitro (peripheral blood, lymphocytes)	Alzheimer’s disease	D	[235]
Malondialdehyde (MDA)	in vitro (erythrocytes)		D	[236]
	in vivo (BAL fluid)	RWE	D	[25]
	in vivo (serum, liver)	Cholesterol	D	[83]
	in vivo (lung)	Acute lung injury (ALI)	D	[208]
	in vitro (U937)	H_2_O_2_	D	[237]
	in vivo (amniotic fluid)			
	in vivo (hippocampus)	Age	D	[229]
	in vitro (HepG2)	Acrylamide	D	[224]
	in vivo (testis)	Deoxynivalenol	D	[226]
	in vivo (serum, liver, kidney)	HgCl_2_	D	[215]
	in vitro (N2a)	FAC	D	[225]
	in vitro (AML-12), in vivo (liver)	Ethanol	D	[218]
	in vivo (liver, kidney)	Thioacetamide	D	[238]
	in vivo (liver)	Bleomycin	D	[233]
	in vitro (peripheral blood, lymphocytes)	Alzheimer’s disease	D	[235]
	in vivo (liver)	CCl_4_	D	[239]
	in vivo (serum, longissimus muscle)		D	[240]
Aspartate Aminotransferase (AST)	in vivo (blood)	D-galactosamine, CCl_4_, LPS	D	[210]
	in vivo (serum)	Cholesterol	D	[83]
	in vivo (serum, liver)	Furosine, Pyralline, 5-Hydroxymethylfurfural	D	[241]
	in vivo (serum, liver, kidney)	HgCl_2_	D	[215]
	in vivo (serum, liver)	Ethanol	D	[218]
	in vivo (serum)	Thioacetamide	D	[238]
	in vivo (blood)	Thioacetamide	D	[195]
	in vivo (serum)	HgCl_2_	D	[215]
	in vivo (serum)	Bleomycin	D	[233]
	in vivo (serum)	LPS	D	[234]
Alanine Aminotransferase (ALT)	in vivo (serum)	HgCl_2_	D	[215]
	in vivo (blood)	Thioacetamide	D	[195]
	in vivo (serum)	Bleomycin	D	[233]
	in vivo (liver)	High-fructose corn syrup	D	[242]
NADPH oxidase (NOX2)	in vivo (hippocampus)	Age	D	[229]
Antioxidants				
Antioxidants	**Study model**	**Inhibitor**	**LF effects**	**Ref.**
Superoxide dismutase (SOD)	in vitro (WBC)		I	[228]
	in vivo (lung)	ALI	I	[208]
	in vivo (hippocampus)	Age	I	[229]
	in vitro (HepG2)	Acrylamide	I	[224]
	in vivo (testis)	Potassium dichromate (PDC)	I	[227]
	in vivo (cortex, hippocampus)	FAC	I	[225]
	in vivo (liver, kidney)	Thioacetamide	I	[238]
	in vivo (liver)	CCl_4_	I	[239]
Catalase (CAT)	in vivo (lung)	ALI	I	[208]
	in vitro (HepG2)	Acrylamide	I	[224]
	in vivo (testis)	PDC	I	[227]
	in vivo (serum, liver, kidney)	HgCl_2_	I	[215]
	in vivo (liver, kidney)	Thioacetamide	I	[238]
	in vivo (liver)	Thioacetamide	I	[195]
	in vivo (serum, longissimus muscle)		I	[240]
Glutathione reductase (GSH)	in vitro (erythrocytes)		I	[236]
	in vivo (serum, liver)	Cholesterol	I	[83]
	in vivo (lung)	ALI	I	[208]
	in vitro (HepG2)	Acrylamide	I	[224]
	in vivo (testis)	PDC	I	[227]
	in vivo (serum, liver, kidney)	HgCl_2_	I	[215]
	in vitro (N2a)	FAC	I	[225]
	in vitro (AML-12) in vivo (liver)	Ethanol	I	[218]
	in vivo (liver)	Bleomycin	I	[233]
	in vitro (peripheral blood, lymphocytes)	Alzheimer’s disease	I	[235]
Glutathione peroxidase (GPX)	in vitro (WBC)		I	[228]
	in vivo (lung)	ALI	I	[208]
	in vivo (testis)	Deoxynivalenol	I	[226]
	in vitro (N2a, SH-SY5Y) in vivo (cortex, hippocampus)	FAC	I	[225]
	in vitro (AML-12) in vivo (liver)	Ethanol	I	[218]
	in vivo (liver)	Bleomycin	I	[233]
	in vivo (liver)	CCl_4_	I	[239]
	in vivo (serum, longissimus muscle)		I	[240]
Total antioxidant status (TAS)	in vivo (amniotic fluid)		I	[237]
Total antioxidant capacity (TAC)	in vivo (serum, liver, kidney)	HgCl_2_	I	[215]
	in vitro (peripheral blood, lymphocytes)	Alzheimer’s disease	I	[235]

D: decrease, I: increase.

**Table 8 biomolecules-15-01174-t008:** Action mechanisms of lactoferrin in host defense.

Actions	Mode of Action	Target Pathogens/Molzecules	LF Effects		Ref.
Antimicrobial action	Iron chelation	Bacteria (G^+^/C^−^), Viruses, Fungi, Parasites	−	Inhibit growth, Prevent infection by blocking binding to host cells	[55,56,57,58,59,63,70,71,73,75,76,77,79,80,81,82,84,85,93,102,104,105,106,107,108,110,114,115,116,118,121,126,129,130,137,139,142,143,145,146,147,148,149,150,151,152,153,154,155,156,157,158]
	Membrane disruption				
	Interaction impairment				
Immune system reaction	Signaling molecules	MAPK, NF-κB	D I	Reduce the expression of pro-inflammatory genes	[178]
	Cytokines	IL-4, IL-10, IL-11, IL-12,			[35,183,185,186,190,191,195,196,199,200,202,209,210]
		IFN-γ, IL-1α, IL-1β, IL-2, IL-6, IL-8, TNF-α, MIF	D		[35,54,74,179,181,182,183,184,185,186,187,188,189,190,191,192,193,194,195,196,197,198,200,201,202,203,204,205,206,207,208,211,212]
Redox regulation	Oxidative stress	ROS, H_2_O_2_, NO, MDA, AST, ALT, NOX2	D I	Protect host cells from damage caused by excessive oxidative stress	[24,25,83,195,197,208,210,213,215,217,218,219,220,222,224,226,228,229,231,232,233,234,235,236,237,238,239,240,241,242]
	Antioxidant activity	SOD, CAT, GSH, GPX, TAS, TAC			[83,195,208,215,218,224,225,227,228,229,233,235,236,237,238,239,240]

D: decrease, I: increase.

## Data Availability

No new data were created or analyzed in this study.

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
