# Peer review of "The Multifaceted Functions of Lactoferrin in Antimicrobial Defense and Inflammation"

_biomolecules, 2025, doi:10.3390/biom15081174_

Round 1

Reviewer 1 Report

Comments and Suggestions for Authors

The manuscript by Jung Won Kim et al. provides a comprehensive review of the role of lactoferrin in inflammation and its antimicrobial properties. In a concise and detailed manner, the review explains the possible uses of lactoferrin, a multifunctional iron-binding glycoprotein that plays a central role in host defence, particularly in protection against infection and tissue injury.

Considering lactoferrin's biological functions, this review emphasises its roles in microbial infections, immune modulation, oxidative stress regulation and inflammation, and highlights its therapeutic potential as a natural agent for managing infectious and inflammatory diseases.

In my opinion, considering the topics covered, this review summarizes correctly the antimicrobial activity and the role in inflammation. However, I have some questions for the authors:

  1. Lines 35–50. It would be useful to include a figure showing the structural organisation of lactoferrin.
  2. similarly, readers would appreciate a schematic figure summarising the different roles of lactoferrin.
  3. Please use italics for the terms “in vitro” and “in vivo” throughout the main text and tables.
  4. In my opinion, it would be interesting to include a paragraph on the potential use of lactoferrin as a drug and bioactive molecule carrier. This would emphasise its potential as a natural carrier that can be directed against specific targets.

Author Response

August 10, 2025

Dear Reviewer,

We appreciate the careful review and constructive suggestions. As you suggested, we have made revisions. Please find enclosed the revised version of the manuscript biomolecules-3758625 entitled “The Multifaceted Functions of Lactoferrin in Antimicrobial Defense and Inflammation”.

We have made extensive revisions and explanations point-by-point as suggested by you. The changes made in the manuscript are written in red. Below, we provide a detailed response to your comments. We hope this improved version of the manuscript, together with our answers below, will adequately address all the inquiries.

Sincerely,

Chaekyun Kim, Ph. D.

Laboratory for Leukocyte Signaling Research,

Department of Pharmacology,

Inha University School of Medicine,

100 Inha-ro, Michuhol-gu, Incheon 22212, Korea

E-mail: chaekyun@inha.ac.kr

Phone: 82-32-860-9874, Fax: 82-32-885-8302

Response to Reviewer's Comments

  1. Lines 35–50. It would be useful to include a figure showing the structural organization of lactoferrin.

R: We have included a figure showing the structural organization of lactoferrin, highlighting its functional roles, as Figure 1.

  1. Similarly, readers would appreciate a schematic figure summarising the different roles of lactoferrin.

R: We have included a figure summarizing the different roles of lactoferrin (Figure 2).

  1. Please use italics for the terms “in vitro” and “in vivo” throughout the main text and tables.

R: We have corrected it.

  1. In my opinion, it would be interesting to include a paragraph on the potential use of lactoferrin as a drug and bioactive molecule carrier. This would emphasise its potential as a natural carrier that can be directed against specific targets.

R: We have included the following paragraphs, including the potential use of lactoferrin as a drug in the Concluding remarks section.

Line 255 ~ 265: Regarding safety and toxicity, Vishwanath-Deutsch et al. reviewed evidence indicating that Lf is well-tolerated and safe in both animal and human studies [251]. Animal studies showed no significant toxicity across safety or tolerability endpoints, with no observed adverse effects even at the highest tested doses. Additionally, no studies have specifically identified increased immunogenicity or allergenicity associated with Lf. Furthermore, Lf is expected to enhance drug bioavailability by encapsulating therapeutic agents and protecting them from degradation. The improved bioavailability may be particularly beneficial for treating intestinal inflammatory disorders. Lf can cross biological barriers, including the blood-brain barrier and intestinal epithelium, making it a promising vehicle for drug delivery. This property is particularly advantageous for targeting neurodegenerative diseases such as Parkinson’s disease. Conjugation of therapeutic agents with Lf can improve their solubility, stability, and targeted delivery for antimicrobial, anti-inflammatory, and chemotherapeutic agents. Therefore, Lf holds promise as a carrier for drugs and bioactive molecules.

Reviewer 2 Report

Comments and Suggestions for Authors

The manuscript thoroughly summarizes lactoferrin’s antimicrobial, antiviral, antifungal, anti-inflammatory, and antioxidant roles with extensive literature support. Additionally, the emphasis on lactoferrin’s role in immune modulation and its potential as a therapeutic candidate adds translational value. Also, the review is well-supported with recent and diverse references. However, several areas require improvement. The following suggestions should be addressed prior to publication.

  1. In the Introduction, the authors should clearly define the research gap that the review intends to address.
  2. The manuscript would benefit from a dedicated section discussing conflicting findings, unresolved questions. Currently, the content is largely descriptive and lacks critical evaluation of contradictory results or emerging controversies in the field.
  3. Several mechanisms, such as iron sequestration and membrane disruption, are repeated across multiple sections, leading to redundancy. It is advisable to summarize overlapping mechanisms in a comparative table to improve clarity and cohesion.
  4. Include at least one schematic figure illustrating lactoferrin’s antimicrobial and immune-modulating pathways to enhance visual understanding of complex interactions.
  5. The manuscript should also feature a brief discussion of relevant clinical trials, including any available safety and efficacy data, to bridge the gap between experimental evidence and clinical application.
  6. It is recommended that the authors explain how specific structural features of lactoferrin (e.g., iron-binding lobes, glycosylation patterns) contribute to its diverse functional roles.
  7. Finally, add a separate section addressing limitations (e.g., bioavailability, stability), challenges (e.g., dosing, safety profiles), and emerging trends or novel therapeutic applications of lactoferrin.

Author Response

August 10, 2025

Dear Reviewer,

We appreciate the careful review and constructive suggestions. As you suggested, we have made revisions. Please find enclosed the revised version of the manuscript biomolecules-3758625 entitled “The Multifaceted Functions of Lactoferrin in Antimicrobial Defense and Inflammation”.

We have made extensive revisions and explanations point-by-point as suggested by you. The changes made in the manuscript are written in red. Below, we provide a detailed response to your comments. We hope this improved version of the manuscript, together with our answers below, will adequately address all the inquiries.

Sincerely,

Chaekyun Kim, Ph. D.

Laboratory for Leukocyte Signaling Research,

Department of Pharmacology,

Inha University School of Medicine,

100 Inha-ro, Michuhol-gu, Incheon 22212, Korea

E-mail: chaekyun@inha.ac.kr

Phone: 82-32-860-9874, Fax: 82-32-885-8302

Response to Reviewer's Comments;

  1. In the Introduction, the authors should clearly define the research gap that the review intends to address.

R: We appreciate your suggestion. We have included the research gap that this review intends to address.

Line 70 ~76: There is extensive evidence supporting Lf’s multifunctional roles in antimicrobial function, immune regulation, and oxidative stress mitigation. Moreover, Lf demonstrates therapeutic potential across a range of infectious and inflammatory conditions. However, a comprehensive understanding of how these diverse biological functions are linked and can be effectively leveraged for clinical application remains limited. This review aims to provide an integrated overview of Lf’s biological functions and to explore its potential as a therapeutic agent. To this end, we outline key microbial species inhibited by Lf and cytokines modulated by its activity, offering a valuable reference to inform future research and the development of Lf-based therapeutic strategies.

  1. The manuscript would benefit from a dedicated section discussing conflicting findings, unresolved questions. Currently, the content is largely descriptive and lacks critical evaluation of contradictory results or emerging controversies in the field.

R: We have added a discussion on conflicting results to the appropriate points of the manuscript.

Line 142 ~149: Interestingly, while most studies report antiviral effects, some findings suggest that Lf may enhance adenovirus infection by promoting viral attachment to epithelial cells, highlighting the virus-specific nature of Lf activity [104]. The authors proposed that Lf facilitates adenovirus infection by binding to the coxsackievirus and adenovirus receptor (CAR), which allows entry into target cells. However, CAR is rarely expressed on the apical side of polarized cells, which initiates infection. Therefore, it seems that adenovirus uses Lf as a bridge to attach to host cells.

Line 190 ~ 195: Lf activates macrophages and enhances TNF-α production [182,183], whereas it suppresses TNF-α production under conditions of chronic endometritis, pregnancy, and lung cancer [198,202,211]. Moreover, Lf differentially regulates the production of cytokines, such as IL-6, IL-10, and TNF-α, depending on the timing of treatment in LPS-induced inflammatory mice [212]. These results suggest that Lf differentially regulates inflammatory cytokines in a context-dependent manner: it promotes pro-inflammatory cytokine production under physiological conditions but suppresses them in pathological or disease states.

  1. Several mechanisms, such as iron sequestration and membrane disruption, are repeated across multiple sections, leading to redundancy. It is advisable to summarize overlapping mechanisms in a comparative table to improve clarity and cohesion.

R: We have added a table summarizing the mechanisms (Table 8).

  1. Include at least one schematic figure illustrating lactoferrin’s antimicrobial and immune-modulating pathways to enhance visual understanding of complex interactions.

R: We have included a schematic figure summarizing the different roles of lactoferrin (Figure 2).

  1. The manuscript should also feature a brief discussion of relevant clinical trials, including any available safety and efficacy data, to bridge the gap between experimental evidence and clinical application.

R: Thank you for your thoughtful suggestion. We have included the following in the Concluding remarks section.

Line 247 ~ 258: Clinical applications of Lf have primarily focused on conditions including anemia, hepatitis C infection, type 2 diabetes, and colorectal polyps. In pregnant women, supplementation with iron-saturated bovine Lf led to increased hemoglobin and total serum iron levels while reducing IL-6 production, indicating both hematologic and anti-inflammatory benefits [203,244-247]. Moreover, the safety and effectiveness of Lf compared to ferrous sulfate treatment have been reported [248]. In patients with chronic hepatitis C, Lf administration resulted in a decrease in HCV viral load and a reduction in serum alanine transaminase levels [117,249]. Furthermore, Lf has been shown to inhibit the growth of adenomatous colorectal polyps, suggesting its potential role in colorectal cancer prevention and as an adjunctive therapy following polyp extraction [250].

Regarding safety and toxicity, Vishwanath-Deutsch et al. reviewed evidence indicating that Lf is well-tolerated and safe in both animal and human studies [251]. Animal studies showed no significant toxicity across safety or tolerability endpoints, with no observed adverse effects even at the highest tested doses. Additionally, no studies have specifically identified increased immunogenicity or allergenicity associated with Lf. 

  1. It is recommended that the authors explain how specific structural features of lactoferrin (e.g., iron-binding lobes, glycosylation patterns) contribute to its diverse functional roles.

R: We have included a figure showing the structural organization of lactoferrin, highlighting its functional roles, as Figure 1.

  1. Finally, add a separate section addressing limitations (e.g., bioavailability, stability), challenges (e.g., dosing, safety profiles), and emerging trends or novel therapeutic applications of lactoferrin.

R: We have included the following paragraphs, including the potential use of lactoferrin as a drug in the Concluding remarks section.

Line 258 ~ 265: Furthermore, Lf is expected to enhance drug bioavailability by encapsulating therapeutic agents and protecting them from degradation. The improved bioavailability may be particularly beneficial for treating intestinal inflammatory disorders. Lf can cross biological barriers, including the blood-brain barrier and intestinal epithelium, making it a promising vehicle for drug delivery. This property is particularly advantageous for targeting neurodegenerative diseases such as Parkinson’s disease. Conjugation of therapeutic agents with Lf can improve their solubility, stability, and targeted delivery for antimicrobial, anti-inflammatory, and chemotherapeutic agents. Therefore, Lf holds promise as a carrier for drugs and bioactive molecules.

Round 2

Reviewer 1 Report

Comments and Suggestions for Authors

In my opinion, the authors have addressed my concerns and requests. Therefore, I consider the revised manuscript acceptable for publication if the editor agrees.